# How Can We Prevent Mother-to-Child Transmission of HTLV-1?

**DOI:** 10.3390/ijms24086961

**Published:** 2023-04-09

**Authors:** Kazuo Itabashi, Tokuo Miyazawa, Kaoru Uchimaru

**Affiliations:** 1Aiseikai-Memorial Ibaraki Welfare and Medical Center, Ibaraki 3100836, Japan; 2Department of Pediatrics, Showa University School of Medicine, Tokyo 1428666, Japan; 3Department of Hematology/Oncology, Research Hospital, The Institute of Medical Science, The University of Tokyo, Tokyo 1088639, Japan; 4Laboratory of Tumor Cell Biology, Department of Computational Biology and Medical Sciences, Graduate School of Frontier Sciences, The University of Tokyo, Tokyo 1088639, Japan

**Keywords:** human T-cell leukemia virus type 1 (HTlV-1), adult T-cell leukemia (ATL), mother-to-child transmission, antenatal screening, prevention, nutritional regimens

## Abstract

The perception of human T-cell leukemia virus type 1 (HTlV-1) infection as a “silent disease” has recently given way to concern that its presence may be having a variety of effects. HTLV-1 is known to cause adult T-cell leukemia (ATL), an aggressive cancer of peripheral CD4 T cells; however, it is also responsible for HTLV-1-associated myelopathy/tropical spastic paraparesis (HAM/TSP). Most patients develop ATL as a result of HTLV-1 mother-to-child transmission. The primary route of mother-to-child transmission is through the mother’s milk. In the absence of effective drug therapy, total artificial nutrition such as exclusive formula feeding is a reliable means of preventing mother-to-child transmission after birth, except for a small percentage of prenatal infections. A recent study found that the rate of mother-to-child transmission with short-term breastfeeding (within 90 days) did not exceed that of total artificial nutrition. Because these preventive measures are in exchange for the benefits of breastfeeding, clinical applications of antiretroviral drugs and immunotherapy with vaccines and neutralizing antibodies are urgently needed.

## 1. Introduction

There are seven known carcinogenic viruses: Epstein–Barr virus (EBV; also known as human herpesvirus 4 (HHV4)), hepatitis B virus (HBV), hepatitis C virus (HCV), human T-cell leukemia/lymphotropic virus type 1 (HTLV-1), human papilloma virus (HPV), Kaposi’s sarcoma herpesvirus (KSHV; also known as human herpesvirus 8 (HHV8)), and Merkel cell polyomavirus (MCV). These viruses are responsible for approximately 15% of all human cancers [1]. Among these viruses, HTLV-1 was the first retrovirus discovered to be carcinogenic [2,3], and the patients showed unique clinical characteristics. HTLV-1 is known to cause adult T-cell leukemia (ATL), an aggressive cancer of peripheral CD4 T cells; however, it is also responsible for HTLV-1-associated myelopathy/tropical spastic paraparesis (HAM/TSP) [4,5]. HTLV-1 carriers have an estimated lifetime risk of 2–5% for the development of ATL [6] and 0.25–1.8% for HAM/TSP [7,8]. Both diseases exhibit serious clinical manifestations and poor prognosis despite therapeutic efforts. In addition to these two diseases, HTLV-1-associated uveitis is widely known [9]. Although ATL and HAM/TSP have been recognized as major outcomes of HTLV-1 infection, they account for less than 10% of the infected population. A recent meta-analysis showed that disease outcomes (morbidity and mortality) for which HTLV-1 is not a necessary diagnostic criterion are clearly associated with HTLV-1 infection (10). This analysis showed a 57% increased risk of early mortality, independent of ATL or HAM/TSP. As ATL and HAM/TSP are present in less than 10% of cases, increased mortality cannot be explained by these two diseases alone [10]. Thus, HTLV-1 infection is likely to affect human health in more ways than currently known and is a global burden [11,12].

Although not ubiquitous, HTLV-1 is found worldwide, and clusters of high endemicity often exist near areas where the virus is rarely present. This could be explained by factors including a possible founder effect, predominance of mother-to-child transmission (MTCT), and cell-to-cell transmission mechanisms [13,14]. The estimated number of HTLV-1 carriers is 5–10 million people across the world [13]. This number may be underestimated because it does not include Russia, China, and India, countries with large populations. The most endemic regions are the southwestern part of Japan, sub-Saharan Africa, South America, the Caribbean region, and foci in the Middle East and Australo-Melanesia regions [13]. HTLV-1-infected individuals are often asymptomatic. Therefore, there is concern about the silent spread of mother-to-child and horizontal transmission.

Most ATL cases are attributed to MTCT [15,16]. The MTCT rate is estimated to be approximately 20% [11], and if we assume a 5% lifetime risk of developing ATL, it is estimated that 25% of cases of MTCT are at a risk of developing ATL. HTLV-1 is a latent virus. Because the host immune system cannot eliminate the virus, HTLV-1 persists in the host and poses a lifelong threat of the development of ATL, HAM/TSP, and other diseases [10,17]. Currently, there is no effective antiretroviral therapy (ART) in clinical use, and the only available measure is the prevention of infection. In this article, we review how HTLV-1 MTCT can be prevented and discuss the challenges in prevention measures.

## 2. Mechanisms of HTLV-1 Transmission

### 2.1. Cell-to-Cell Transmission

HTLV-1 virions are rarely detected in the extracellular environment [18]. Thus, HTLV-1 infection is believed to spread predominantly through direct cell-to-cell contact. Cell-to-cell transmission may enhance the multiplicity of infection and evade the host immune responses. It also aids rapid viral replication kinetics by directing virus assembly and budding to sites of cell-to-cell contact [19,20]. Although in vivo evidence has not been established yet, in vitro studies have suggested that HTLV-1 cell-to-cell transmission may occur through viral synapses [21], conduits [22], biofilm-like structures [23], and extracellular vesicles [24]. Recently, Hiyoshi et al. [25] reported that the host factor M-Sec, which induces membrane protrusion and establishes intercellular conduits, plays an important role in efficient viral infection. These modes seem to be favorable for the virus to escape immune elimination (HTLV-1-specific T-cell unresponsiveness) and efficiently reach contacted cells, resulting in increased proviral load (PVL) [26]. HTLV-1 predominantly infects CD4+ T cells via cellular receptors such as heparin sulfate (HS) proteoglycans and neuropilin 1 (NRP-1), which help in initial binding to the cell and glucose transporter 1 (GLUT1) [27,28,29,30,31].

Recent studies have shown that cell-free HTLV-1 can infect certain types of cells rather than being poorly infectious as previously thought [27]. In vitro studies have shown that HTLV-1 infection of T cells via dendritic cells (DCs) can occur in two different ways: In cis-infection, after infecting DCs, de novo produced HTLV-1 is transferred to T cells [32]. In trans-infection, uninfected DCs capture the virus produced by infected T cells and transfer it to T cells before becoming infected [32,33,34].

Because DCs, monocytes, epithelial cells, macrophages, and B cells express these receptors, they can also infect each other in individuals with HTLV-1 infection [27,35]. CD4+ T cells are the primary targets of HTLV-1 infection in vivo [36]. In addition, HTLV-1 proviral DNA can be detected in CD8+ T cells [37], DCs [38], plasmacytoid dendritic cells [39], and monocytes, including macrophages [35,40], albeit to a lesser extent.

### 2.2. HTLV-1 Life Cycle

Infected lymphocytes transmit HTLV-1 through intercellular contact with target cells, and viral components, including the single-stranded RNA genome of HTLV-1, are transferred to target cells through these junctions [41]. HTLV-1 genomic RNA (gRNA) is reverse-transcribed in the cytoplasm of target cells, resulting in double-stranded DNA of size 9 kb, which is inserted into the host genome in the target cell nucleus to form a provirus. The position at which the double-stranded DNA is inserted is not completely random. HTLV-1 is preferentially incorporated into characteristic regions; however, the underlying mechanism is currently unknown [42,43]. The provirus is transcribed by RNA polymerase II in the cell and is modified post-transcriptionally. Both full-length and spliced viral mRNAs are transported from the nucleus to the cytoplasm. Viral proteins are then translated by the translation machinery of the host cell, and Gag, Gag-Pol, and Env proteins are transported to the plasma membrane along with two copies of HTLV-1 gRNA. Immature viral particles are formed from these viral proteins and gRNAs, which release from the cell surface. Subsequently, viral proteases act on immature viral particles to form mature viral particles with infectious potential (see Martin’s review [44] for a detailed description of this process).

### 2.3. HTLV-1 Replication

According to previous studies, immediately after infecting the cell, the HTLV-1 virus spreads from cell to cell. Later, during the chronic infection phase, the virus survives through clonal expansion as a provirus, which is incorporated into the host cell genome and replicates as the infected cells divide [27,45]. Replication of HTLV-1 occurs via (i) an infection cycle involving viral budding and infection of new target cells and (ii) mitosis of cells harboring an integrated provirus [46]. During HTLV-1 integration into the host genome, the 5’ and 3’ ends of HTLV-1 are duplicated to form long terminal repeats (LTRs). These regions constitute the promoter regions as transcription factor binding sites. The proviral genome comprises the structural genes gag, pol, and env flanked by LTR at both ends. The genome also contains the pX region, which has four partially overlapping open reading frames encoding p12, p13, p30, Rex, and Tax, which are regulatory or accessory proteins. The viral genes are transcribed from the 5’ LTR. HTLV-1 also expresses a minus-stranded RNA that encodes HTLV-1 bZIP factor (HBZ), a basic leucine zipper factor protein. *HBZ* is the only gene that is encoded in the antisense strand and is transcribed from the 3’ LTR. The HTLV-1 genome has the potential to express multiple products using various strategies, such as frameshifting and alternative mRNA splicing.

*Tax* and *Rex* are essential for viral replication. *Tax* promotes viral mRNA synthesis by transactivating the HTLV-1 promoter located in the 5’ LTR. Tax acts in a coordinated manner on various intracellular targets during cell transformation and is involved in immortalization, cell proliferation, and leukemogenesis. On the other hand, Tax is a major target antigen recognized by cytotoxic T lymphocytes (CTLs) [47]. Therefore, for HTLV-1-infected cells to survive, *Tax* expression must be tightly regulated to evade host immune surveillance. Tax expression is normally suppressed to escape CTLs, but at the same time Tax is transiently expressed to maintain and expand HTLV-1-infected cells [48]. The *HBZ* gene is the only HTLV-1 gene present in all infected individuals. Unlike Tax, HBZ is always expressed but is less immunogenic, and thus more likely to escape CTL clearance. Furthermore, HBZ may suppress the effects of Tax, leading to survival of infected cells and oncogenesis [49]. *Rex* regulates the synthesis of structural proteins at the post-transcriptional level [50]. The accessory proteins p12/p8, p13, and p30 are important for viral infectivity and persistence in vivo but are not essential for viral replication in vitro [51,52,53].

## 3. Modes of HTLV-1 Transmission

There are two modes of HTLV-1 transmission: horizontal infection and antenatal or postnatal MTCT [15,54]. In 2013, there were an estimated 1780 pregnant carriers in Japan [55]. In addition, the MTCT rate in a recent prospective cohort study in Japan was observed as 4.5% (95% confidence interval (C.I.) 2.6–7.4%) [56]. Based on these data, the number of new mother-to-child infections is estimated to be 70 (95% C.I. 41–115) per year. The number of new horizontal infections in Japan is estimated to be approximately 4000 per year, which is far larger than the number of new infections caused by MTCT.

### 3.1. Horizontal Transmission

The main sources of horizontal infection are sexual intercourse, blood transfusion, and parenteral transmission via contaminated needles. According to the WHO Technical Report, 23 countries have implemented mandatory screening for HTLV-1 antibodies in all donated blood samples. However, despite being mandatory, HTLV-1 antibody screening is not always performed during blood donations by the same person in these countries [11]. Since donor blood screening for HTLV-1 infection is always performed at the time of blood collection [57], horizontal infection occurs mainly through sexual transmission in Japan [58]. Organ transplantation from an HTLV-1 carrier has also been identified as a cause of horizontal HTLV-1 infection, and the addition of HTLV-1 antibody testing to donor testing has been advocated [59].

The Miyazaki Cohort Study examined heterosexual HTLV-1 transmission in 534 couples over a five-year period from 1984 to 1989. This study showed that the infection rate was 3.9 times higher when the carrier spouse was male [60]. Satake et al. evaluated 3,375,821 repeat blood donors aged 16–69 years for new HTLV-1 infection over a 4.5-year period. Their results were as follows [58]: (i) at least 4000 adolescents and adults were estimated to be newly infected each year, (ii) the incidence density was significantly higher in women (6.88 per 100,000 person-years; 95% C.I. 6.17–7.66) than in men (2.29 per 100,000 person-years; 95% C.I. 1.99–2.62; *p* < 0.0001), (iii) the highest number of newly infected individuals were males in their 60s and females in their 50s, regardless of endemic area, (iv) a higher number of males in their 20s were newly infected in metropolitan areas (non-endemic areas) than in non-endemic areas. As new infections in adolescence and adulthood are primarily caused by sexual transmission in Japan, reports advocate the importance of preventing horizontal transmission from a public health perspective.

Factors related to sexual intercourse include non-use of contraceptives, numerous partners, and male-to-male intercourse [61]. Kaplan et al. found that high PVL and length of relationship played a role in viral transmission from male carriers to non-carrier women [7]. A higher PVL tends to be associated with HAM/TSP [62], ATL [63], HTLV-1-associated infectious dermatitis [64], and HTLV-1 uveitis [65]. In addition, PVL tends to be higher in patients co-infected with *Strongyloides stercoralis* than in the others [66]. Sexual transmission occurs more efficiently from men to women than women to men and might be enhanced by sexually transmitted diseases that cause ulcers and result in mucosal ruptures, such as syphilis, herpes simplex type 2 (HSV-2), and chancroid [67]. Other sexually transmitted diseases may result in the recruitment of inflammatory cells and increase the risk of HTLV-1 acquisition and transmission [61].

### 3.2. Mother-to-Child Transmission

The main reason for the focus on MTCT of HTLV-1 is that most ATL cases originate from MTCT [64], and ATL rarely develops in individuals infected during adulthood [6].

#### 3.2.1. Transmission Routes of MTCT

The Nagasaki ATL Prevention Program found that exclusive formula feeding (ExFF) markedly reduced the HTLV-1 MTCT rate from 20.3% to 2.5% [68]. Accumulating evidence has shown that the HTLV-1 MTCT rate in children who were exclusively fed infant formula was significantly lower than in children who were breastfed for an extended period [68,69,70,71]. Therefore, the primary route of MTCT is through breastfeeding. However, MTCT has been observed in a small percentage of children (approximately 2.5–6.7%) exclusively fed infant formula [56,68,71]. This suggests the possibility of antenatal MTCT [54].

##### Antenatal Transmission

The presumed pathways for antenatal MTCT are intrauterine and the birth canal. A recent study showed that trophoblasts in pregnant carriers are highly susceptible to HTLV-1, suggesting that intrauterine infection may occur due to impairment of the blood–placental barrier [72]. However, there is little clinical evidence for intrauterine ascending infection, intrapartum infection due to exposure to contaminated maternal blood, or intrauterine infection [51].

##### Transmission through Breastfeeding

It is unclear which infected cells in breast milk are transmitted to the infant and how MTCT is established. It has been noted that viral uptake during lactation may occur in the tonsillar mucosa, the intestinal mucosa, or both sites [73], while postnatal infection is thought to occur when infected cells in ingested breast milk enter the infant’s digestive tract [74,75]. The number of leukocytes in breast milk decreases to 0–2% of the total cell count within a few weeks of lactation. In addition to leukocytes, many other cell types are present in breast milk, including mammary luminal epithelial cells, mammary gland cells, and stem/progenitor breast cells, which vary with lactation period, maternal conditions, and infant feeding [76]. HTLV-1 MTCT has been thought to be primarily mediated by CD4+ T cells, but several studies have suggested that mammary epithelial cells and macrophages may be involved in the persistence and spread of HTLV-1 infection from the carrier mother [77,78,79]. However, it remains unclear which cells present in breast milk are the main players in breast milk infection. The process from the contact of infected cells with the mucosa to the spread of infection in the submucosal tissue has been described in detail in several reviews [27,46,80], and the following process has been postulated by Carpenter et al. [46]: (i) bilions incorporated into vesicles migrate from the apical surface of epithelial cells to the basal surface of the epithelial cell [73], (ii) newly produced virions are released from the basal surface of infected epithelial cells [80], (iii) HTLV-1-infected cells are bypassed through the injured mucosa [81], and (iv) macrophages pass through the epithelium, as seen with HIV [82]. The process by which infected cells in breast milk enter the infant’s gastrointestinal tract and establish infection is not yet fully understood.

#### 3.2.2. Risk Factors Associated with MTCT

Since the 1980s, it has been widely recognized that extended breastfeeding is a risk factor for MTCT, and as discussed below, avoidance of breastfeeding is an important measure for preventing MTCT [71,83]. However, the involvement of other factors should be considered when testing pregnant women, particularly in countries or regions where maternal HTLV-1 antibody screening is not routinely performed [84]. Furthermore, even if HTLV-1 screening tests are performed on all pregnant women, as in Japan [83], intervention measures considering the risk factors are desirable to minimize avoidance of breastfeeding. The risk factors for MTCT reported to date are shown in Figure 1. However, sensitivity and specificity of these factors, except the duration of breastfeeding, in predicting MTCT have not been sufficiently studied. Plancoulaine et al. detected chromosome 6q27 as the dominant gene that predisposes individuals to HTLV-I infection based on a large genetic epidemiological study on an HTLV-1 endemic population of African descent living in French Guiana [85,86,87].

There has long been an interest in whether the presence or transfer of antibodies in breast milk plays an effective role in preventing MTCT. Moreover, it has been reported that pregnant women infected with HTLV-1 have significantly increased levels of anti-HTLV-1 antibodies, although their PVL did not change during pregnancy [88]. This results in the transmission of more antibodies to the fetus through the placenta during pregnancy. This report is consistent with the hypothesis that infection may be prevented in fetuses and early postnatal infants.

Rosadas et al. measured anti-HTLV-1/2 IgG antibodies and PVL in paired blood and breast milk from HTLV-1/2-positive mothers and reported that HTLV-1 PVL and IgG binding ratios were similar in plasma and breast milk; however, the anti-HTLV-1/2 IgG antibody titer in plasma was approximately 1000 times higher than that in breast milk [89]. After delivery, HTLV PVL increased in the mother’s blood [90]. Given the antepartum and postpartum changes in PVL and antibodies in infected mothers, as well as the lower antibody levels in breast milk, MTCT prevention with short-term breastfeeding (discussed below) may be less likely to involve IgG antibodies in breast milk. One reason for the increased risk of MTCT with prolonged breastfeeding may be related to lower levels of transitional antibodies during infancy and increased cumulative intake of infected cells ingested through breast milk. High maternal PVL has also been identified as a risk factor for MTCT [91,92]. This was also reflected in elevated maternal antibody titers [93].

Other risk factors for carrier mothers include HAM/TSP complications [94], co-infection with *Strongyloides stercoralis* [94], ≥2 previous children with HTLV-1 infection [91], high PVL in breast milk [95], and human leukocyte antigen (HLA) class I type concordance between mother and child via breastfeeding [96]. Furthermore, in mothers with untested HTLV-1 antibodies from endemic areas, a lack of effective intervention may result in MTCT.

Substances present in breast milk, such as tumor growth factor (TGF)-β and lactoferrin, which are abundant in colostrum [92,97], promote HTLV-I replication [98,99]. Furthermore, lactoferrin expression has been shown to be elevated during HTLV-1 infection [100]. However, the levels of these components are not constant during lactation and vary from person to person. Therefore, it is unclear how they affect MTCT.

## 4. Strategies to Prevent HTLV-1 MTCT

Theoretical strategies to prevent the MTCT of HTLV-1 include avoidance of breastfeeding, reduction in infected cells in breast milk, and administration of vaccines, neutralizing antibodies, and antiretroviral drugs. These strategies are discussed in the following sections. Other important strategies include promoting the use of condoms to prevent transmission to uninfected women from male carriers. Furthermore, it is essential to disseminate knowledge about HTLV-1 infection not only to medical providers and health administrators but also to the general public.

### 4.1. Prevention of MTCT through Nutritional Regimens

Several nutritional regimens have been proposed to prevent the MTCT of HTLV-1 (Table 1) [54]. However, some methods provide limited evidence. Previous epidemiological and animal studies have shown that most HTLV-1 MTCT occurs through breast milk containing infected cells. Therefore, ExFF, which intercepts breast milk containing infected cells, is theoretically the most reliable method for postnatal prevention. As mentioned above, a follow-up study by the ATL Prevention Program (APP), which started in 1987 in the Nagasaki Prefecture, showed that ExFF reduced the rate of MTCT to approximately 1/10 of that after long-term breastfeeding (≥6 months) [68]. However, it has been suggested that the longer a carrier mother breastfeeds her infant, the higher the MTCT rate [91].

In Japan, methods such as limiting the duration of breastfeeding to three to six months or inactivating infected cells by freezing and thawing procedures (frozen–thawed breast milk feeding; FTBMF) have been proposed as alternatives to ExFF for carrier mothers who wish to breastfeed their babies [83]. In the Kagoshima Prefecture, an endemic area of Japan, short-term breastfeeding (STBF) has historically been promoted if mothers wish to breastfeed, and over 60% of mothers have opted for STBF [101]. This indicates that a significant number of HTLV-1 carrier mothers wished to breastfeed their infants. However, because the effectiveness of these interventions in preventing MTCT is based on small observational studies rather than randomized controlled trials, sufficient evidence is lacking.

In a recent technical report on HTLV-1, the WHO recommends that “available data should be further analyzed to better define the risk of HTLV-1 transmission associated with specific duration of breastfeeding, balanced with the risks of other adverse health outcomes that may result from reduced breastfeeding” [102]. In this context, Itabashi et al. conducted a prospective multicenter cohort study involving HTLV-1 carrier pregnant women and their infants as part of the Health, Labor, and Welfare Science Research Program in Japan to determine the rate of MTCT by ExFF, STBF, and FTBMF [56]. Miyazawa et al. reported findings through a systematic review that integrated the results of the cohort study and previous studies [103].

#### 4.1.1. Exclusive Formula Feeding (ExFF)

A meta-analysis of 12 studies by Rosadas et al. in 2022 showed that the risk of MTCT with breastfeeding (of any duration) was approximately four times higher than that with ExFF [84], supporting the effectiveness of avoiding breastfeeding for the prevention of infection. However, ExFF lacks the various positive effects of breastfeeding, such as nutritional and immunological benefits, long-term disease prevention, economic efficiency, promotion of mother–infant bonding, and promotion of maternal recovery after delivery. Many HTLV-1 carrier mothers are concerned that they cannot form mother–infant bonds because they cannot breastfeed their babies [104].

According to a review article by Millen et al., avoidance of breastfeeding is not an option in resource-limited areas or populations with few infected individuals [105]. In particular, in developing countries with high morbidity and mortality rates of serious gastrointestinal and other infections due to poor sanitation, which do not provide a stable supply of formula, baby bottles, and clean water, the advantages of the immunological benefits of breast milk may outweigh the disadvantages of the MTCT of HTLV-1. Therefore, the recommended level of breastfeeding avoidance to prevent HTLV-1 MTCT should be considered based on each local situation.

#### 4.1.2. Short-Term Breastfeeding (STBF)

Although the precise mechanism of MTCT prevention by STBF is unknown, it is assumed to be due to the transplacental transfer of neutralizing antibodies from the mother to infant during pregnancy. The antibodies remain in the infant for several months after birth and may prevent MTCT during the first few months of life. The period of exposure to the infected cells is short, and the cumulative number of infected cells entering the digestive tract is small.

In the Nagasaki Prefecture, the duration of STBF has been set at six months or less since the late 1980s. During this period, the MTCT rate was 2.4% (23/962) for ExFF, while it was significantly higher for STBF (≤6 months) at 8.3% (14/169) [106]. Since the late 1990s, when the duration of STBF was changed to three months or less, the MTCT rate was observed to be 3.7% (8/218) for ExFF versus 2.8% (1/36) for STBF, with no statistical difference between the two [106]. In the Kagoshima Prefecture, between 1986 and 2006, the MTCT rate for ExFF was 4.8% (16/331), whereas that for STBF (≤3 months) was 1.6% (2/126) [107]. Based on these results, the recommended period of STBF is less than three months (less than 90 days after birth) in Japan [54].

In a Japanese prospective cohort study by Itabashi et al., the intention-to-treat (ITT) analysis showed that the MTCT rates for STBF (less than 90 days) and ExFF were 2.3% (4/172 children born to carrier mothers) and 6.4% (7/110), respectively, with no statistically significant difference between the two groups [56]. Among 172 mother–infant pairs who chose STBF, 33.5% were still breastfeeding at three months of age and 7.8% at six months, and the approximate formula suggests that 18.2% were still breastfeeding at 4 months of age [56]. Thus, even if a mother chooses STBF, it is difficult for her to terminate breastfeeding and make the transition to ExFF within ≤3 months (90 days) of age of the infant. In addition, there is a concern that prolonged breastfeeding may increase the risk of MTCT.

A 2021 systematic review included a meta-analysis of the risk of MTCT of STBF ≤ 3 months and STBF ≤ 6 months compared with that of ExFF [103]. The meta-analysis integrated five retrospective studies and the cohort study by Itabashi et al.; comparing STBF ≤ 3 months (including <90 days) with ExFF found no statistical difference in the risk of MTCT between the two groups (pooled risk ratio (RR): 0.72, 95% CI: 0.30–1.77) (Figure 2) [103]. In contrast, a meta-analysis integrating five retrospective studies and comparing STBF ≤ 6 months and ExFF showed that STBF ≤ 6 months was associated with an approximately 3-fold higher risk of MTCT than that of ExFF (pooled RR: 2.91, 95% CI: 1.69–5.03) (Figure 3) [103]. Although there was no statistical difference in the MTCT rates between STBF ≤ 3 months and ExFF, Rosadas et al. documented that all studies included in the meta-analysis were observational studies in Japan [84].

#### 4.1.3. Frozen–Thawed Breast Milk Feeding (FTBMF)

It is speculated that freeze–thaw treatment of breast milk destroys infected cells, thereby inactivating its infectivity in the infant [54]. Specifically, the expressed breast milk is frozen in a home freezer at −20°C or lower for at least 24 h, thawed, and fed to the infant. Milk expression, freezing, and thawing are necessary and time-consuming processes. In a Japanese prospective cohort study, only 19 of 313 mothers opted for FTBMF, and MTCT was confirmed in one infant [56]. A meta-analysis integrating three prospective observational studies, including the cohort study by Itabashi et al., found no difference in the risk of MTCT between ExFF and FTBMF (pooled RR: 1.14, 95% CI: 0.20–6.50) [103]. However, the number of cases analyzed was small, the subjects were limited to Japan, and the duration of FTBMF was not constant, and included cases of a short duration (2–6 months) [108,109]. Therefore, it may be premature to conclude that FTBMF is an effective intervention to prevent MTCT. However, FTBMF is routinely administered to preterm infants born at less than 32 weeks of gestation who are at risk for infection, necrotizing enterocolitis, and related deaths [110]. Thus, FTBMF would outweigh the risk of HTLV-1 MTCT while in the neonatal intensive care unit (NICU) for such infants.

FTBMF requires several work processes. If an infant born to an infected mother is admitted to the NICU, the mother’s work involves expressing and freezing breast milk, and then bringing the frozen breast milk to the NICU. However, if not admitted to the NICU, two additional processes are required: thawing the frozen breast milk and transferring it to a bottle. It is difficult to repeat a series of work processes on a daily basis.

#### 4.1.4. Milk Pasteurization and Banked Human Milk

When newborn infants cannot be fed with their mother’s milk, such as preterm infants admitted to the NICU, human milk donated to human milk banks is an important resource for supporting their health. According to international guidelines [111], milk is pasteurized using the Holder method (62.5 °C for 30 min). According to a systematic review conducted by Pitino et al., all viruses studied, except parvoviruses, are susceptible to thermal killing [112]. Unfortunately, this review did not report any studies on HTLV-1. Yamato et al. reported that heat treatment (56°C for 30 min) eliminated HTLV-1 activity in an in vitro study [113], but no subsequent clinical studies have been conducted to date. Theoretically, this is sufficient to suppress transmission of infection through breastfeeding; however, further studies are required to clarify this issue.

Banked human milk should be screened for maternal HTLV-1 infections [114]. Theoretically, banked human milk could have the same preventive effect as ExFF in infants born to HTLV-1 carriers. However, while banked human milk may provide some health benefits for infants and children [115], it is unlikely to reduce carrier mothers’ anxiety and/or impairment of mother–child bonding. This method would be available assuming that resources are abundant and a breast milk banking system exists; however, clinical studies must be conducted before this can be performed.

#### 4.1.5. Mixed Feeding

The method of supplementing the deficiency with infant formula in the case of decreased breast milk secretion is called mixed feeding. Some carrier mothers intentionally choose mixed feeding immediately after birth to reduce the amount of breast milk ingested by their infants, thus reducing the amount of virus transferred to them. However, the effect of this approach on MTCT is unknown. In recent studies, the rate of MTCT of HIV was extremely high (approximately 20%) compared to normal breastfeeding or infant formula feeding [116]. Mixed feeding may cause gastrointestinal mucosal injury or dysbiosis, resulting in changes in intestinal permeability [117]. It is possible that the same concept can be applied to the MTCT of HTLV-1. However, there is a lack of evidence recommending mixed nutrition immediately after birth.

### 4.2. Prevention Methods Other Than Nutritional Regimens

To prevent the MTCT of HIV, antiretroviral prophylaxis, cesarean section, and avoidance of breastfeeding are now sentinel events in resource-rich countries [118]. These are expected to be effective in preventing the MTCT of HTLV-1, which belongs to the same Retroviridae family.

Bittencourt et al. reported that when elective cesarean sections were performed in 81% of 41 HTLV-1 carrier pregnant women who opted for ExFF, no MTCT was observed in any of the 41 infants [119]. Although elective cesarean sections are expected to be effective in minimizing an infant’s exposure to mother’s blood containing infected cells, no high-quality studies have been conducted to date, and no evidence exists to support elective cesarean sections [15,71,84]. Conclusively, carrier pregnant women should not be generally indicated for cesarean section, as it may increase the risk of complications for mothers and children.

To date, no clinical trials have been conducted on ART during pregnancy, although in vitro studies have suggested efficacy of ART [71]. In a case series published in the United Kingdom in 2021, zidovudine was administered to four mothers who developed ATL during pregnancy and to their babies. The authors reported that MTCT was observed in one of the four mothers, but the outcomes of the other three were unknown because of the short follow-up period [120]. Since there have been no studies on asymptomatic carriers who have not developed ATL, further investigation is warranted. Despite promising in vitro data, clinical data on the efficacy of antiretroviral drugs in preventing the MTCT of HTLV-1 are scarce [84].

Previous animal experimental and pilot studies have suggested that immunotherapy, such as neutralizing antibodies and vaccination with the HTLV-1 gene product, may protect against infection [121,122,123,124]. The ideal candidates and methods of inoculation remain to be elucidated. Furthermore, the correlates of the immune response have not yet been elucidated. Even if clinically effective vaccines and neutralizing antibodies are developed, they may be targeted first to those at high risk of sexual transmission, followed by the prevention of MTCT (see the review article by Ratner [125]).

## 5. Screening Program and Strategies for Prevention of MTCT in Japan

### 5.1. Background

Introduction of an HTLV-1 antibody screening program for all pregnant women remains controversial [88,91,126]. HTLV-1 antibody screening tests for all pregnant women are currently unavailable in countries except Japan. A nationwide antenatal HTLV-1 antibody screening program was implemented in Japan since 2010 owing to the following reasons: (i) HTLV-1 carriers are spread throughout Japan by internal population movement from endemic areas to non-endemic areas [127]; (ii) more than 4000 adolescents and adults are newly infected through sexual contact [58]; (iii) no effective drug treatment has been developed against this virus [128]; (iv) reduction in the number of these children would also contribute to a reduction in horizontal sources of transmission.

### 5.2. Screening Program in Practice

HTLV-1 antibody screening is usually performed within 30 weeks of gestation, allowing carriers sufficient time to obtain more information from their healthcare providers before delivery and select the appropriate feeding regimen for their infants. Pregnant women with positive screening results undergo confirmatory antibody testing using an algorithm (Figure 4) [54,129]. If a pregnant woman is determined to be a carrier, the healthcare provider will explain the risks of MTCT and preventive measures to the extent possible before delivery. If the mother does not have strong concerns about the risks of HTLV-1 associated diseases and interventions for MTCT, infant and child health examinations are performed on the same schedule (at 1, 3–4, 7–10, 18, and 36 months) as for infants born to non-carrier pregnant women. Testing for HTLV-1 antibodies at the age of three years to assess MTCT is recommended, but not mandatory [83].

### 5.3. Nutritional Regimens in Japan

Since 2017, the Japanese nutrition protocol for the prevention of postnatal MTCT via breast milk has changed from the three previous options of ExFF, STBF, and FTBMF to ExFF as the first choice with the most reliable preventive effect [83]. Based on the results of a recent cohort study and meta-analysis by Itabashi and Miyazawa [56,103], it was concluded that the MTCT rate for STBF would not exceed the risk of MTCT for ExFF unless the duration of breastfeeding does not exceed 90 days after birth and that adequate maternal support by a medical care provider is a precondition for ensuring this. Sufficient evidence to prove the effectiveness of FTBMF has not yet been obtained; therefore, it is not recommended [130]. Medical providers should not uniformly recommend ExFF to mothers from the perspective of MTCT prevention, but should fully explain the advantages and disadvantages of each nutritional regimen from the perspective of pregnancy, delivery, and childcare and provide shared decision-making support so that mothers can make their own decisions about nutritional methods, including STBF and other nutritional regimens (Table 1) [54].

### 5.4. Issues of Nationwide Antenatal Screening Program in Japan

#### 5.4.1. Support for Carrier Mothers

More than 10 years have passed since the screening test for all pregnant women was introduced in Japan. However, carrier mothers were not satisfied with the status quo. Their main opinions were as follows: (i) it is difficult to say that medical providers adequately support carrier mothers’ choice of nutritional regimen, and (ii) there are no medical facilities close by where mothers can discuss their concerns about the onset of the disease or their children’s infection, and they do not know whom to consult. This might be due to limited experience and insufficient knowledge of obstetricians and pediatricians about HTLV-1 infection, as well as lack of collaboration among obstetricians, pediatricians, hematologists or neurologists, and local government officials. Therefore, establishment of a consultation and support system for carrier mothers and their families based on local medical resources, along with public awareness of HTLV-1 infection, is an urgent issue.

#### 5.4.2. Selection of Nutrition Regimen Considering Risk Factors

It is unclear whether it is appropriate to select ExFF or STBF without considering risk factors for MTCT. Deprivation of long-term breastfeeding in infants at very low risk for MTCT may impact their future health [131]. In addition, there are concerns about the impact of the selection of a nutrition regimen on mothers’ parenting behavior and mother–infant bonding [132]. Future studies should accumulate data on infants born to carrier mothers to determine the association between MTCT and its risk factors and to minimize the avoidance of breastfeeding.

#### 5.4.3. Follow-Up of a Child Infected via MTCT

Although HAM/TSP is generally considered an adult manifestation of HTLV-1, the possibility of early-onset HAM/TSP via MTCT has long been reported, mainly in South America [133,134,135]. Yoshida et al. reported a case of childhood HAM/TSP in Japan in 1993 [136]; however, only a few cases have been reported since then.

Dermatological lesions, such as infectious dermatitis, atopic dermatitis, seborrheic dermatitis, acquired ichthyosis, candidiasis, palmar erythema, dermatophytosis, crusted scabies, and folliculitis decalvans, may be associated with HTLV-1 infection [137]. Cutaneous involvement in an apparently asymptomatic carrier has been considered a premonitory indication for the future development of either ATL or HAM/TSP [137]. As PVL may slightly fluctuate in asymptomatic children, measurement of PVL on a regular basis may not be of much clinical significance [138]. However, as early-onset HAM/TSP and ATL may be associated with a variety of skin lesions in addition to infectious dermatitis [133,134], serological tests and PVL measurements may be useful in children with known MTCT in endemic areas [138]. Children with known MTCT and suspected of having neurological abnormalities, such as weakness, muscle stiffness, spasm, gait disturbance, and abnormal urination, should be considered for PVL measurements. Skin lesions are also observed in such cases. The association between skin lesions and early-onset HAM/TSP in children in Japan has rarely been discussed. Since atopic dermatitis and seborrheic eczema occur frequently in infants in Japan, regardless of HTLV-1 infection, pediatricians are not concerned about their appearance in HTLV-1-infected children via MTCT. Longitudinal follow-up is needed to determine whether the relationship between skin lesions and premature HTLV-1-related disease in infected mother and child pairs in Japan differs from that in South America.

If MTCT is obvious, parents should consider at what age the child will be informed and who will inform the child of this fact. Furthermore, if the child is anxious, counseling may be necessary.

## 6. Conclusions

The perception of HTLV-1 infection as a “silent disease” has recently given way to concern that its presence may be having a variety of effects. Therefore, measures to prevent mother-to-child and horizontal transmissions are becoming increasingly important. Currently, no antiretroviral drugs or immunotherapies can be used clinically. More than 90% of MTCT cases involve trans-breastfeeding; therefore, the main preventive measures are avoidance of breastfeeding or reduction in infected cells in breast milk. Our study indicated that the MTCT rate of STBF within 90 days of birth in infants born to carrier mothers did not exceed that of ExFF. However, it is estimated that approximately 20% of mothers who choose STBF are unable to discontinue it by 90 days; therefore, adequate support from healthcare providers is essential. ExFF and STBF are available only in resource-rich areas with good sanitation. On the other hand, breastfeeding has various advantages. Accurate prediction based on risk factors for MTCT may curb more over-intervention cases for infants born to carrier mothers in resource-rich countries and reduce cases where the benefits of breastfeeding are traded off. However, in countries with limited medical resources, ExFF may not be a realistic option, particularly because it is directly associated with increased infant mortality. If antiretroviral drugs and immunotherapy, such as vaccines and neutralizing antibodies, are introduced in the future, it is expected that they may contribute to the prevention of MTCT after birth without compromising the advantages of breastfeeding and may even be useful for prenatal prevention of infection.

## Figures and Tables

**Figure 1 ijms-24-06961-f001:**
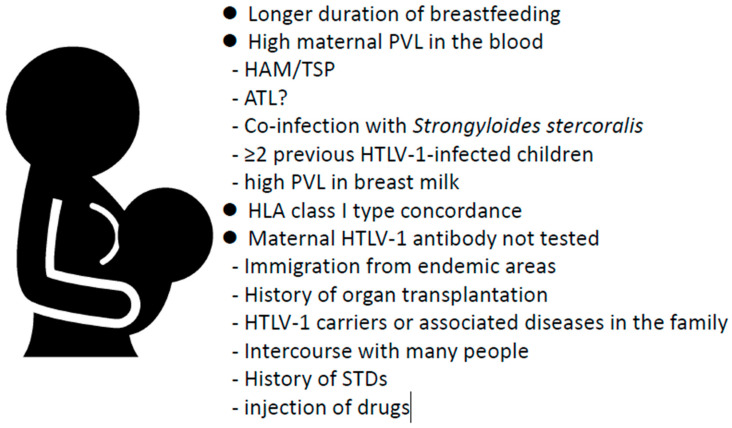
Risk factors associated with development of HTLV-1 mother-to-child transmission. Risk factors for mother-to-child transmission are broadly classified as long-term breastfeeding, high PVL in carrier mothers, HLA class type 1 concordance between mother and child, and mothers with untested HTLV-1 antibodies. HLA, human leukocyte antigen.

**Figure 2 ijms-24-06961-f002:**
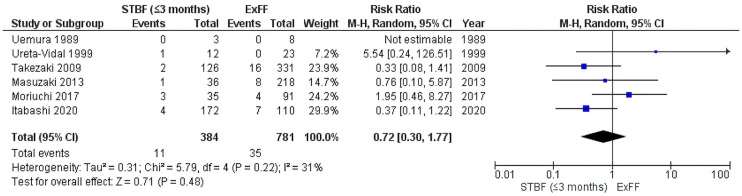
Forest plot of the risk ratios of HTLV-1 MTCT in the STBF ≤ 3 months group compared with that of the ExFF group. There is no statistical difference in the risk of MTCT between the two groups (pooled risk ratio (RR): 0.72, 95% CI: 0.30–1.77). Abbreviations: STBF, short-term breastfeeding; ExFF, exclusive formula feeding; M–H, Mantel–Haenszel; CI, confidence interval; RR, risk ratio; MTCT, mother-to-child transmission; events, number of cases with mother-to-child transmission; total, number of children born to carrier mothers; weight, influence of studies on overall meta-analysis. The figure is reproduced from Miyazawa et al. [103].

**Figure 3 ijms-24-06961-f003:**
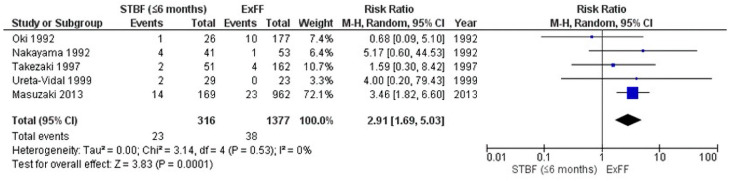
Forest plot of the risk ratios of HTLV-1 MTCT in the STBF ≤ 6 months group compared with that of the ExFF group. Comparing STBF ≤ 6 months and ExFF showed that STBF ≤ 6 months was associated with an approximately 3-fold higher risk of MTCT than that of ExFF (pooled RR: 2.91, 95% CI: 1.69–5.03). Abbreviations: STBF, short-term breastfeeding; ExFF, exclusive formula feeding; M–H, Mantel–Haenszel; CI, confidence interval; RR, risk ratio; MTCT, mother-to-child transmission; events, number of cases with mother-to-child transmission; total, number of children born to carrier mothers; weight, influence of studies on overall meta-analysis. The figure is reproduced from Miyazawa et al. [103].

**Figure 4 ijms-24-06961-f004:**
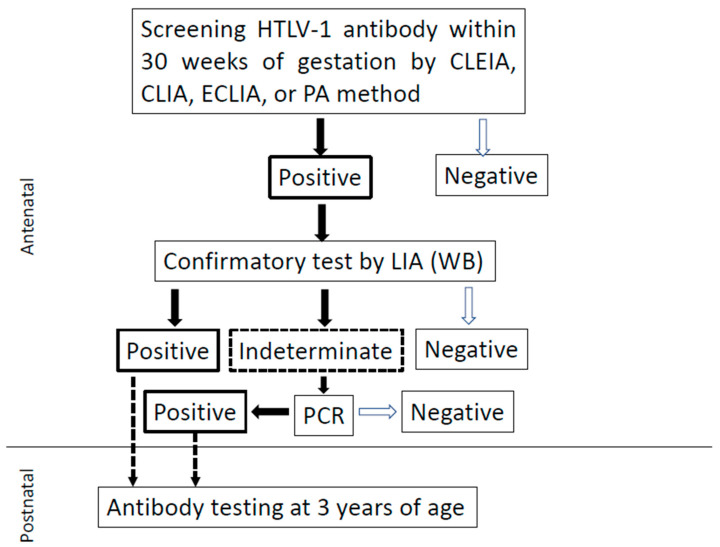
Algorithm for diagnosing HTLV-1 infection in Japan. Currently, no confirmatory tests using Western blotting are conducted in Japan. Flowchart for identifying HTLV-1 carriers among pregnant women. CLEIA, chemiluminescent enzyme immunoassay; CLIA, chemiluminescent immunoassay; ECLIA, electrochemiluminescent immunoassay; PA, particle agglutination; WB, Western blotting; LIA, line immunoassay; PCR, polymerase chain reaction. The figure is reproduced from Itabashi et al. [54]. For further details, please refer to Okuma et al. [129].

**Table 1 ijms-24-06961-t001:** Effectiveness of feeding regimens in peventing mother-to-child transmission and their limitations.

Nutritional Regimens	Effectiveness on MTCT	Comments
Exclusive infant formula feeding (ExFF)	Widely used and well evaluated to block MTCT through breast milk	Prevents about 95% or more of MTCT No benefits from breastfeeding Concerns about increased risk of postpartum depression and impaired mother–child bonding
Short-term breastfeeding (≤3 months)	No apparent difference in the MTCT prevention effect (vs. ExFF) Majority of studies in Japan	Acquisition of some benefits of breastfeeding Approximately 18% of children exceed 4 months of breastfeeding Need to provide adequate support for weaning No data on the preventive effect of postpartum depression or impairment of mother–child bonding
Short-term breastfeeding (≤6 months)	Approximately three times increased risk of MTCT (vs. ExFF)	Better to avoid this regimen
Frozen–thawed breast milk feeding	No apparent difference in the MTCT prevention effect (vs. ExFF) Only three small case studies in Japan, with little confidence in preventive effects	Time-consuming Considered for use in infants admitted in the NICU No data on the preventive effect of postpartum depression or impairment of mother–child bonding
Mixed feeding	Unknown effectiveness of MTCT prevention due to lack of data (vs. ExFF)	Concerns about increased risk of MTCT due to damage to the intestinal mucosa Better to avoid this regimen
Banked human milk pasteurization	No data available, but expected to be as effective as ExFF in preventing MTCT	No use of breast milk from untested HTLV-1 donors No data on the preventive effect of postpartum depression or impairment of mother–child bonding

Note: It should be noted that ~5% of antenatal infections cannot be avoided regardless of which nutritional regimen is chosen. MTCT, mother-to-child transmission; NICU, neonatal intensive care unit. The table is reproduced from Itabashi et al. [54] with some modifications.

## Data Availability

The raw data supporting the conclusions of this article will be made available by the authors, without undue reservation, to any qualified researcher. The data presented in this study are available on request from the corresponding author.

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
