# Peer review of "How Can We Prevent Mother-to-Child Transmission of HTLV-1?"

_ijms, 2023, doi:10.3390/ijms24086961_

Round 1

Reviewer 1 Report

In the manuscript by Itabashi, Miyazawa and Uchimaru, the authors review the current knowledge and public policy on Mother-to-Child Transmission of HTLV-1 and suggest steps moving forward. The review is logically laid out and nicely covers the topic. However, minoe revisions are needed as many of the references are either incorrect or incomplete.

Section 2. Mechanisms of HTLV-1 transmission

Line 72, the authors should cite Sato et al Virology 1992; 186, 712-724 and Jolly Virology 2011; 411(2): 251–259 for the rapid kinetics of viral spread through cell-to-cell transmission.

Line 74, the authors incorrectly cite Malbec et al for conduits transmission of HTLV-1. Van Prooyen et al PNAS, 2010; 107, 20738-43, which was the first report of p8 inducing conduits for viral transmission.

Line 75, the authors incorrectly cite Nakamura et al for biofilm-like structures, The first report of HTLV-1 transmission through bio-films was reported by Pais-Correia et al Nat Med 2010; 16, 83-89.

Line 88, the authors should include Assil et al PLoS Pathog 2019; 15 to transmission by DCs.

Line 93, the authors should include de Castro-Amarante et al J Virol. 2015; 90, 2195-207 for HTLV-1 infection of macrophages and monocytes.

Line 141, reference Edwards et al Viruses 2011; 3, 861-85 is a review discussing p12/p8, p13 and p30. Gross et al Viruses 2016; 8, 74 is a review that discusses cell-to-cell transmission but not the role of p13 or p30. The authors could cite Omsland et al Retrovirology 2020; 17, 11 and Moles et al Retrovirology 2019; 16, 42.

The authors should check that all appropriate references have been cited for all sections of the review.

Author Response

Thank you for your detailed peer review.

-Line 72, the authors should cite Sato et al Virology 1992; 186, 712-724 and Jolly Virology 2011; 411(2): 251–259 for the rapid kinetics of viral spread through cell-to-cell transmission.

    I have made the following corrections with the addition of the cited references pointed out. "It also aids rapid viral replication kinetics by directing virus assembly and budding to sites of cell-to-cell contact (19,20). "

In line 74, changed the citation from Malbec's paper to van Prooyen's paper.

In line 75, changed the reference from Nakamura's paper to Pais-Correia's paper.

Line 88, added Assil's paper to the references.

Line 93, added de Castro-Amarante's paper to the references.

Line 141, removed the Gross paper and added the Omasland paper and the Moles paper as references.

The appropriateness of references other than those pointed out was also reexamined.

Reviewer 2 Report

The present paper is a review in an interesting topic such as prevention in HTLV-1 mother-to-child transmission.  In the absence of effective drug therapy, total artificial nutrition such as exclusive formula feeding is a reliable means of preventing mother-to-child transmission after birth. Although the manuscript is focused on the region of Japan, its contribution is very important since this topic is currently little addressed in the scientific community. It is correctly written and the methodology used is correct. The review is very complete and the conclusions are very interesting. In addition, the review compares the different types of feeding of babies with respect to HTLV infection.

In addition, this infection is usually unknown to health professionals, so I consider it important to disclose the results of different investigations, in order to implement preventive measures for the transmission of HTLV-1 from mother to child.

The tables and figures are correct and easy to view, suitable for a review article. Regarding the bibliography, it is correct, however citations 103 and 104 must be written in English, since they are in Japanese. This manuscript is very interesting and correctly and comprehensively addresses a problem considered a neglected disease, despite being a public health problem in regions where this infection is endemic, such as Japan.

The authors also make a brief description of the transmission routes of the HTLV-1 virus, a particular mechanism of this virus, different from other viruses of the same family. For the above reasons, I consider that the review is very interesting and will be a great contribution to your magazine. Kind regards

Author Response

Thank you for your detailed peer review.
We have removed the Japanese part of the reference and corrected it.